# Development of Measurement Method for Temperature and Velocity Field with Optical Fiber Sensor

**DOI:** 10.3390/s23031627

**Published:** 2023-02-02

**Authors:** Masashi Sekine, Masahiro Furuya

**Affiliations:** 1Nuclear Regulation Authority, 1-9-9 Roppongi, Minato-ku, Tokyo 106-8450, Japan; 2Faculty of Science and Engineering, Waseda University, 3-4-1 Okubo, Shinjuku-ku, Tokyo 169-8555, Japan

**Keywords:** optical fiber thermometer, hot wire anemometer, velocity measurement, temperature measurement, distribution measurement

## Abstract

We have developed a new method for measuring temperature and velocity at a high spatial resolution (minimum 2.56 mm pitch along an optical fiber). The developed method uses the same principle as a hot wire anemometer, where the velocity perpendicular to an optical fiber is estimated as a function of the cooling curve of a gold-coated layer on the optical fiber Joule-heated intermittently. The developed optical fiber sensor demonstrated the ability to acquire a transient velocity profile in airflow experiments with high repeatability and accuracy. This paper describes optical fiber-based velocity measurement in the velocity range of approximately 0–7 m/s with an error of approximately 10% compared to a hot wire anemometer and a new method for simultaneous temperature and velocity measurements. Applicability to velocity distribution measurements and seconds transient velocity changes are also described.

## 1. Introduction

Thermal hydraulics is the phenomenon whereby the temperature and flow fields interact with each other. Examples of the strong coupling between temperature and flow fields include natural convection flow and heat transfer. The specific applications are air conditioning systems in buildings [1], thermal storage tanks [2,3], and passive safety systems in nuclear power plants [4]. Therefore, the measuring of temperature and velocity fields is crucial to understanding the thermal hydraulics in these applications. 

Numerous measurement methods have been developed. Thermocouples for temperature and hot wire anemometers [5] for flow velocity are widely used because they are simple and accurate. On the other hand, immersing these sensors in the measurement domain disturbs the temperature and flow fields, resulting in only a few selected points being measured [6]. Two optical methods are combined to measure temperature and flow fields: Hanwook Park [7] and Stig Grafsrønningen et al. [8] used PIV (Particle Image Velocimetry) and Laser Induced Fluorescence with phosphor tracer particles, and In Mei Sou et al. [9] used PIV and Thermography. Although their optical methods explored temperature and velocity fields, measurement calibration was complex due to significant changes in the refractive index and for optical visibility through windows, three-dimensional shapes, and large temperature gradients [8]. In addition, when tracer particles were mixed into the measurement, the tracking of the particles became excessive. When the circulation force was weak, the tracer might have settled out due to the long time measuring and natural convection. Another measurement method is distribution measurement using optical fibers. Lomperski et al. [10] measured the temperature distribution, although the velocity distribution measurement has not been thoroughly investigated. The simultaneous measurement of temperature and velocity has not been investigated.

We have developed a new temperature and velocity measurement with a high temporal and spatial resolution to overcome these drawbacks using an optical fiber thermometer. The new method can measure three-dimensional temperature and flow velocity distributions even in closed systems without significantly disturbing them.

Section 2 describes the temperature and velocity measurement methods using optical fibers. Section 3 validates the measurement performance using the airflow experiment results. Section 4 describes the condition setting of this measurement method. Finally, Section 5 discusses an applied example of the new method developed in this study.

## 2. Measurement Method

This section describes the newly developed measurement method. The temperature was measured by an optical fiber measurement method. The flow velocity was measured by the temperature change, similar to with a hot wire anemometer. Simultaneous temperature and velocity measurements could be obtained by combining the above two measurements.

Figure 1 shows the schematic of the temperature and velocity measurement devices. For the flow velocity measurement, the following modifications were made to the standard optical fiber thermometer system: (1) coating the optical fiber with gold, and (2) adding devices for pulsed Joule heating of the gold-coated fiber (see the red-dashed boxes in Figure 1). 

### 2.1. Temperature Measurement Method

In this study, the Optical Distributed Sensor Interrogator B (ODiSI-B) from Luna Innovation Inc. was used as the optical fiber measurement device. The measurement device consisted of a tunable laser device, a detector, and an optical fiber sensor, as shown in Figure 1. The frequency shift amount Δf of the Rayleigh backscattered light was detected by ODiSI-B based on the optical frequency domain reflectometry method [11]. The measurement performance of ODiSI-B can measure the amount of frequency shift at a measurement frequency of 100 Hz and a measurement interval of 2.56 mm. The sensor part was calibrated for position and set to 2 m. The fiber diameter was approximately 0.16 mm (+0.002/−0.006).

The frequency shift amount Δf is a function of the temperature difference ΔT and the strain ε of the optical fiber from a reference state. The relationship is given by the following equation [6,10],
(1)$$\Delta f=K_{T}\Delta T+K_{\varepsilon }\varepsilon ,$$
where $K_{T}$ and $K_{\varepsilon }$ are proportional constants, which are determined in the calibration process. If the fiber strain is kept unchanged, the temperature can be estimated from Equation (1) and Δf. Since the values of Δf are obtained at nearly continuous spatial points along the fiber, the temperature can also be measured at the same spatial points.

Lomperski et al. [10] and Sang et al. [12] demonstrated the ability to measure temperature distributions by using optical fibers under high radiation conditions in spent fuel pools of nuclear reactor facilities. Kim et al. [13] demonstrated the applicability of the measurement method at the operating range of 30–70 °C. Weathered et al. [14] and Arora et al. [15] applied the measurement method at high temperatures (up to 600 °C) in sodium and molten salt, respectively. 

### 2.2. Velocity Measurement Method

Velocity across an optical fiber is estimated based on the principle of hot wire anemometry [16]. The gold-coated optical fiber was electrically heated for velocity measurement by a DC power supply with a function generator signal. The temperature of the optical fiber was measured as described in the previous section. Figure 2 shows the temperature change of the optical fiber when pulsed Joule heating was applied to the fiber. The temperature rose from the ambient fluid temperature $T_{f}$ during the heating period and returned to $T_{f}$ during the non-heating period. 

If the heat conduction in the optical fiber is neglected, the temperature decay process during the non-heating period is expressed by the following energy conservation equation:(2)$$\pi r^{2}\cdot \rho C_{p}\left(\frac{dT}{dt}\right)=-2\pi r\cdot h\left(T-T_{f}\right),$$
where h is the heat transfer coefficient the between ambient fluid and the optical fiber, $\rho ,\;C_{p},T,r$ are the density, specific heat capacity, temperature, and radius of the optical fiber, respectively. The general solution of Equation (2) is presented by:(3)$$T=T_{f}+\Delta Te^{-\frac{t}{\tau }},$$
where ΔT is a constant that depends on the applied heat, and
(4)$$\tau =\frac{r\rho C_{p}}{2h}.$$

Since h depends on the flow velocity, τ also depends on the flow velocity from Equation (4). Thus, the flow velocity can be estimated from the τ value, which is obtained by fitting the measured temperature with Equation (3).

### 2.3. Temperature and Velocity Simultaneous Measurement Method

As shown in Section 2.2, the flow velocity was estimated by measuring the temperature change during the temperature decay process after pulsed Joule heating. After the temperature decay process, the temperature approached the ambient fluid temperature $T_{f}$, which is the temperature measurement of our method. Thus, one temperature value and one velocity value were measured per pulse of Joule heating.

## 3. Calibration and Measurement Performance

This section describes the calibration for the temperature and velocity measurement methods presented in Section 2 and the measurement performance.

### 3.1. Temperature Measurement

The values of the constants in Equation (1) depend on the type of fiber and the temperature range. Calibration experiments were conducted by measuring the water temperature in which the gold-coated optical fiber was immersed to determine the value of $K_{T}$ by comparing it with the reference temperature obtained from the Type-K thermocouple (measurement error of ±2.5 °C). The temperature varied in the range of 20–90 °C. As a result, $K_{T}=-1.5$ was obtained, and thus the temperature was estimated by the measured Δf and the following equation,
(5)$$T=T_{0}-0.67\times \Delta f,$$
where $T_{0}$ is the base temperature.

Lomperski et al. evaluated the measurement error of optical fiber thermometers [6,10,15]. The measurement error of the optical fiber thermometer was reported to be ±3.38 °C [15]. In our experiments, the measurement error was also found to be $\pm \sqrt{2.5^{2}+2.8^{2}}=\pm 3.8\;℃$. Here, the measurement error of the reference thermocouple was ±2.5 °C and that of the gold-coated fiber was ±2.8 °C.

### 3.2. Velocity Measurement

Figure 3 shows the experimental setup for the velocity calibration. The gold-coated optical fiber sensor was installed in a 0.1 m diameter pipe. A honeycomb structure was installed at the inlet, and a small fan (SANYO DENKI CO., LTD., Tokyo, Japan) was installed at the outlet to stabilize the airflow. The hot wire anemometer (KANOMAX Inc., Shimizu Suita City, Japan) was used to measure the wind velocity as a reference, and the measurement error was greater than ±3% of the displayed value or 0.02 m/s.

A small fan was used to generate a steady flow for the velocity calibration, and the flow velocity was measured with the hot wire anemometer. At the same time, the time constant τ of the temperature decay process after Joule heating was measured as described in Section 2.2. Figure 4 shows the relationship between the reference velocity $U_{ref}$ measured by the hot wire anemometer and the time constant τ. For the experimental range shown in Table 1, the flow velocity U is accurately approximated by the following expression for powers of τ,
(6)$$U=0.022\tau^{-2.56}$$

The above multiplier relationship can be understood from the fact that Equation (3) gives $\tau \propto h^{-1}\propto Nu^{-1}$ and the Nu number is given by the power of the Re number proportional to the flow velocity. For example, the Zukauskas model [17] for flow around a cylinder shows $Nu\propto Re^{0.4}\left(Re\propto U\right)$, giving $U\propto \tau^{-0.25}$, which is consistent with Equation (6).

Figure 5 shows the results of the comparison between the flow velocity $U_{OF}\;$measured with the optical fiber using Equation (6), and the reference velocity $U_{ref}$ measured with the hot wire anemometer. The green line is the standard deviation, and the flow velocity can be measured by the optical fiber with approximately 10% deviation from the hot wire anemometer. From the above, the temperature decay process after heating depends on the wind velocity, and the prospect of velocity–current measurement using τ was obtained.

### 3.3. Simultaneous Temperature and Velocity Measurements

The simultaneous measurement method described in Section 2.3 was applied to the airflow experiment shown in Figure 3. The reference temperature was provided by the same K-type thermocouple used in the temperature calibration, and the reference velocity was provided by the hot wire anemometer used in the velocity calibration. Regarding experimental conditions, the electric power was 21 kW/m^2^ with heating for 0.2 s, with the output turned on every 2 s. The experiment started when the fan started, and the wind velocity stabilized at 1 m/s. As shown in Figure 6, the temperature and velocity measured once per pulse matched the reference values from the thermocouple and the hot wire anemometer. The response time was 2 s, sufficient for the measurement prospect.

The measurement errors of the simultaneous measurements are summarized in Table 2. Although the velocity measurement error was the same as the value obtained from the velocity calibration, the temperature measurement error was smaller than the value obtained from the temperature calibration. This is because the temperature in the simultaneous measurements was evaluated by fitting the temperature change during the temperature decay process.

## 4. Discussion

This section discusses the setting of the measurement parameters and the effect of wind velocity on the fiber.

### 4.1. Setting of Measurement Parameters

It was necessary to set the power and the heating duration time of the Joule heating applied to the fiber to perform the velocity measurement described in Section 2.2. From the point of view of minimizing the effect on the fluid to be measured, it was desirable to use low power and a short time for the Joule heating. On the other hand, from the point of view of maintaining the accuracy of the fit when determining τ, it was desirable to increase the power or heating duration time of the Joule heating to increase the amount of heat applied to the fiber and to increase the temperature response.

Figure 7 shows the fiber’s temperature response as the power varied between 7, 11, 21, 32, and 42 kW/m^2^ with a heating duration of 0.2 s for the Joule heating. The smaller the power applied to the fiber, the smaller the temperature response to pulse heating and the larger the temperature noise effect. As shown in Table 3, the coefficient of determination R^2^ for obtaining τ was degraded under low power conditions (7 kW/m^2^), so it was set to 11 kW/m^2^ or higher in the experiments in this paper. Since the fiber temperature returns to the ambient temperature faster at lower power, a lower power should be selected when the interval between pulse heating, i.e., the interval between velocity measurements, is small.

In this paper, to confirm the temperature response to pulse heating with high accuracy, the heating duration was set to 0.2 s, which was sufficiently longer than the 0.01 s interval between temperature measurements of the optical fiber. However, from the point of view of shortening the interval between velocity measurements, the heating duration should be shorter.

### 4.2. Effect of Wind Velocity on Fiber Measurement

Equation (1) indicates that the frequency shift of the optical fiber is affected by stress. However, the temperature calibration equation in Equation (5) was calculated under the assumption that the optical fiber is not under stress. Therefore, we checked the effect of the stress received from the airflow using the experimental apparatus shown in Figure 3. The amount of frequency shift when the wind velocity across the optical fiber was varied from 0.0 (stagnant) to 15.0 m/s without heating is shown in Figure 8. The frequency shift was less than 1 GHz (equivalent to 0.67 K) regardless of the flow velocity. This indicates that the stress caused by the wind velocity did not affect the temperature measurement.

## 5. Application Example of Optic Fiber Velocity Measurement System

This section introduces the characteristics of using optical fiber: distribution measurements and time variation of velocity distribution.

### 5.1. Measurement of Wind Velocity Distribution

The calibration equation obtained from a single measurement point on the optical fiber was applied to the entire fiber to measure the velocity distribution. Three measurements were taken each time at wind velocities of 1 m/s, 2 m/s, and 3 m/s in the experimental setup shown in Figure 3. A range of 70 mm was measured because the area near the electrode could not be accurately measured. Figure 9 shows the wind velocity and hot wire anemometer with the optical fiber. The optical fiber could measure velocities at 2.56 mm intervals, with a tendency for higher velocities in the center of the cylinder and slower velocities near the walls. As a trend in wind velocity, there was a slight variation in the optical fiber measurements as the velocity increased. However, compared to the three hot wire anemometers, the measurement was within ±5%, which is less than the ±10% velocity measurement error of the optical fiber.

From the above, the optical fiber velocity meter could measure the velocity distribution within the measurement error of a hot wire anemometer in the measurement range. This is an advantage not found in conventional measuring instruments. On the other hand, the measurement error of this instrument tended to be larger than that of the hot wire velocimeters. This can be attributed to several factors. The main factors considered were the uncertainty in the temperature conversion equation of thermocouples, the uncertainty in calculating the decay time constant, the uncertainty in the velocity calibration equation, and the uncertainty in the gold layer thickness.

### 5.2. Measurement of Wind Velocity Transients

Figure 10 shows the change in velocity distribution when the fan changed the wind direction from the left side to the right side of the optical fiber in 10 s using the experimental airflow apparatus and the conditions shown in Figure 3 in Section 3.2. When the fan was fixed at the center position of the fiber using a hot wire anemometer, the wind velocity was 2.5 m/s. On the other hand, the velocity distribution using the optical fiber was smaller than 2.5 m/s overall because the fan was moved in the wind direction. However, the velocity distribution was observed to move from left to right.

From the above, the optical fiber anemometer could capture velocity changes on the order of seconds. Therefore, we qualitatively obtained the prospect that the anemometer can be applied as a measurement device for transient changes.

## 6. Conclusions

Using an optical fiber thermometer, a new high-resolution, multi-point temperature and velocity measurement method was developed. The scope of application of optical fiber velocimeters and their applicability to distribution, temperature, and velocity measurements were investigated. For the velocity measurement of optical fibers, we applied the principle of hot wire anemometry to temporarily increase the temperature of an optical fiber by pulsed Joule heating of a gold-coated optical fiber. Then, we investigated the velocity measurement method from the temperature time series data of the optical fiber. As a result, the velocity measurement method focusing on the temperature decay process was more effective in the experimental range of 0.2–6.8 m/s, with an error margin of approximately 10% compared to hot wire velocimetry. These results showed that the velocity measurement method focusing on the temperature decay process was adequate for the present experimental range.

The optical fiber velocity measurement method focusing on the temperature decay process confirmed that simultaneous temperature and velocity measurement could be performed every 2 s. Furthermore, the velocity distribution could be measured at 2.56 mm intervals. As a transient change in the velocity distribution, it could be qualitatively captured for wind velocities changing every second, giving us the prospect of applying this measurement system to transient changes.

Future research will focus on its applicability to different fluids, velocity and temperature range limits, and transient temperature and velocity changes.

## Figures and Tables

**Figure 1 sensors-23-01627-f001:**
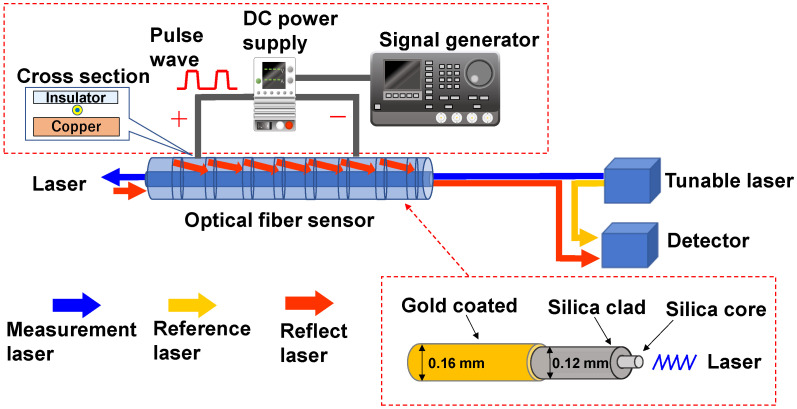
Schematic of the temperature and velocity measurement devices with the gold-coated optical fiber.

**Figure 2 sensors-23-01627-f002:**
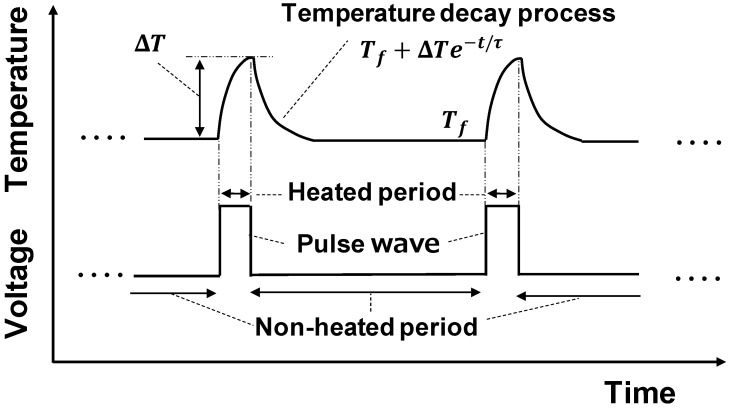
Temperature transients of optical fiber in response to applied Joule-heating pulses.

**Figure 3 sensors-23-01627-f003:**
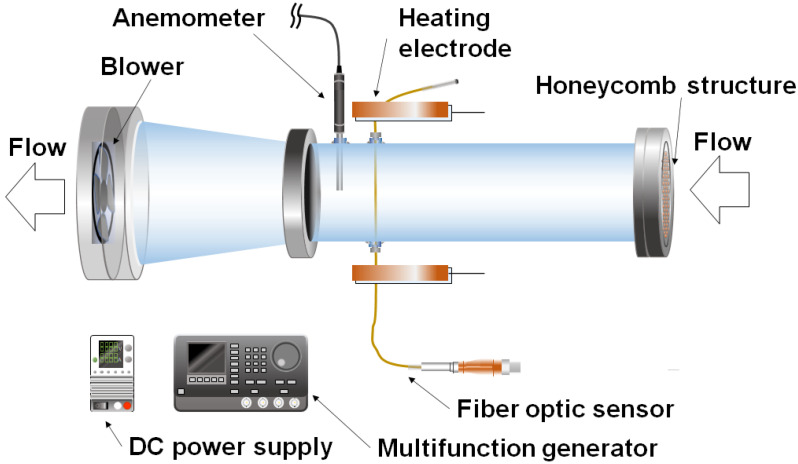
Velocity calibration of the fiber optic sensor in a wind tunnel.

**Figure 4 sensors-23-01627-f004:**
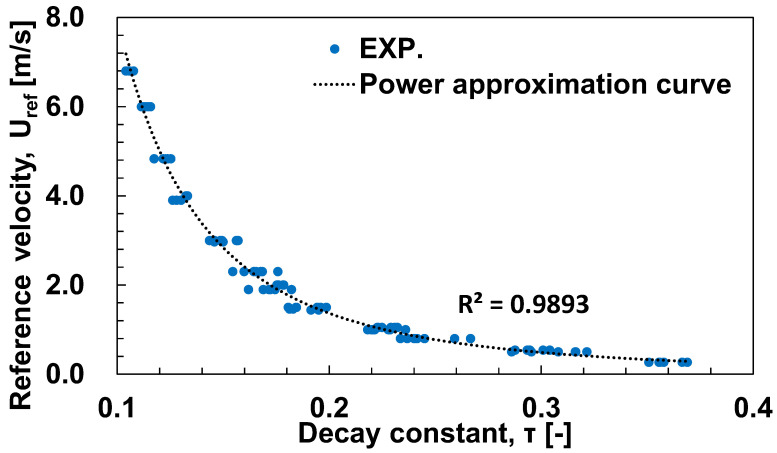
Decay time constant (τ) of temperature transients as a function of reference velocity ($U_{ref}$).

**Figure 5 sensors-23-01627-f005:**
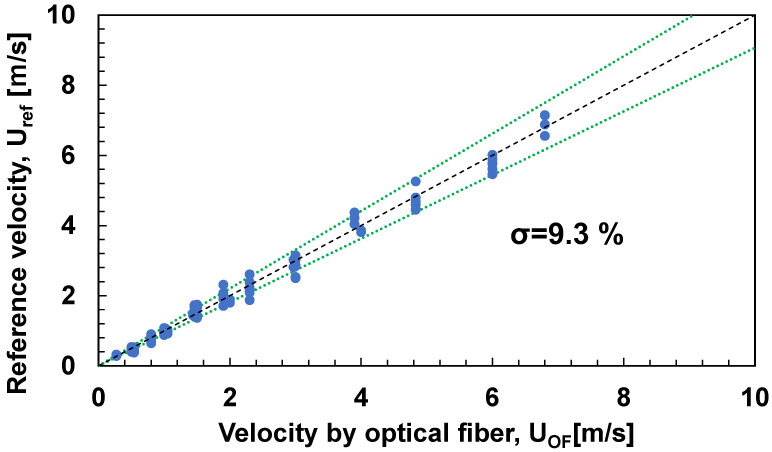
Velocity by optical fiber ($U_{OF}$) against reference velocity ($U_{ref}$).

**Figure 6 sensors-23-01627-f006:**
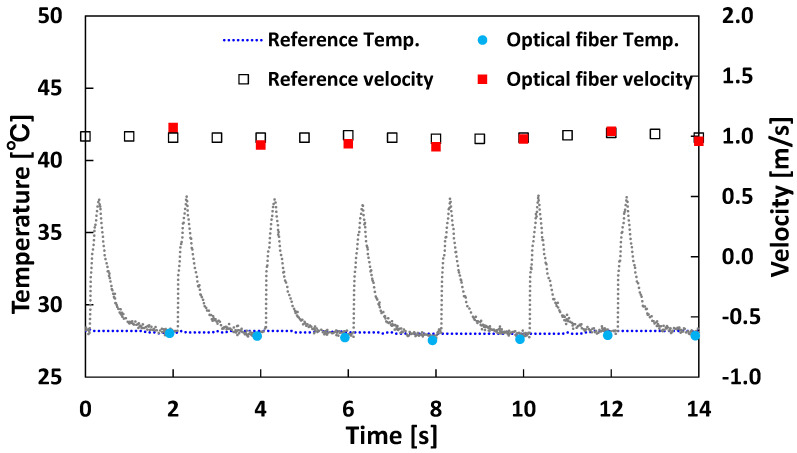
Temperature time series data during temperature and velocity measurements.

**Figure 7 sensors-23-01627-f007:**
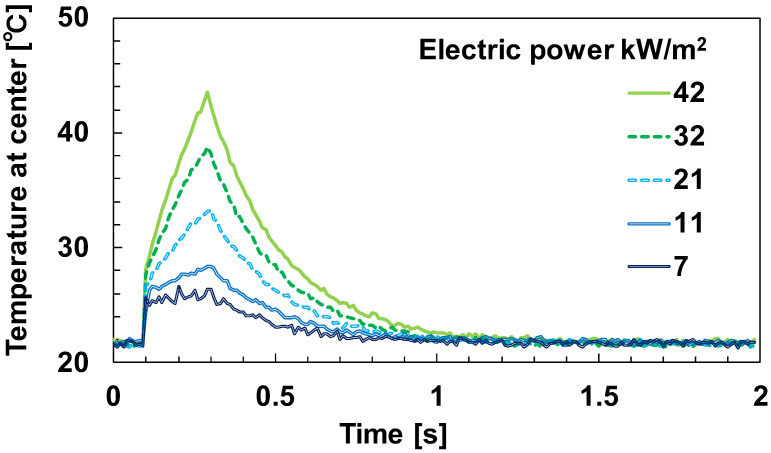
Time trace of temperature with 1.0 m/s and heating for the first 0.2 s in reference to the heating power.

**Figure 8 sensors-23-01627-f008:**
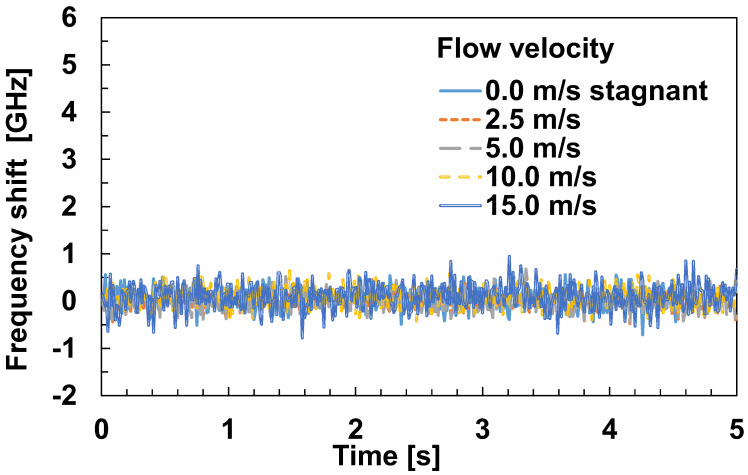
Effects of wind velocity on the frequency shift.

**Figure 9 sensors-23-01627-f009:**
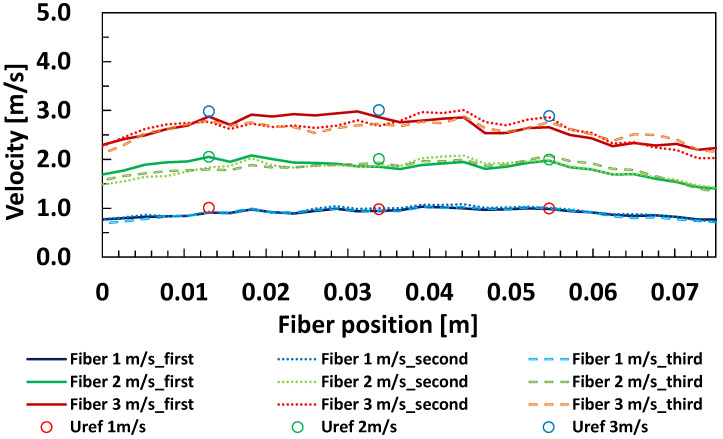
Effect of wind velocity on velocity distribution along the optical fiber.

**Figure 10 sensors-23-01627-f010:**
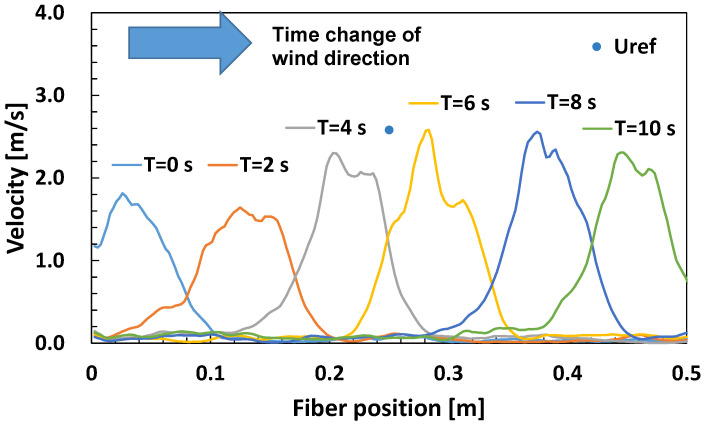
Velocity distribution along the optical fiber in reference to time as snapshots.

**Table 1 sensors-23-01627-t001:** Test parameters and their ranges.

Item	Unit	Range
Electric power	kW/m^2^ *	11, 21, 32, 42
The waveform of electrical output	−	Square wave
Heating duration time	s	0.2
Wind velocity	m/s	0.2–6.8
Fluid temperature	°C	20–29
Measurement frequency	samples/s	100

*: Heat flux is normalized by dividing W from the DC source by the surface area of the fiber.

**Table 2 sensors-23-01627-t002:** Measurement errors of measuring devices.

Device	Unit	Range
Type-K thermocouples	°C	±2.5
Velocity with anemometer	m/s	±3% or ±0.02
Temperature with optical fiber	°C	±3.9
Velocity with optical fiber	m/s	±10%

**Table 3 sensors-23-01627-t003:** R^2^ value per power to fiber.

Electric Power [kW/m^2^]	R^2^
42	0.999
32	0.999
21	0.998
11	0.991
7	0.978

R^2^**:** The coefficient of determination manifests how well the approximation explains the variation in the data. It is defined as the ratio of the sum of squares of the regression to the sum of squares.

## Data Availability

Not applicable.

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
