# Peer review of "Development of Measurement Method for Temperature and Velocity Field with Optical Fiber Sensor"

_sensors, 2023, doi:10.3390/s23031627_

Round 1

Reviewer 1 Report

This paper proposes a new temperature and velocity measurement method at high spatial resolution (minimum 2.56 mm interval along an optical fiber). The application scope of fiber optic velocimeters and their applicability to distribution, temperature, and velocity measurements was investigated. The velocity measurement method focusing on the temperature decay process was more effective in the 0.26.8 m/s experimental range. This is valuable work, but it needs to be revised appropriately. Here are some comments and questions:

1. Only the experimental range of 0.2 ~ 6.8 m/s is mentioned in this paper. What are the measured results in other ranges?

2. Graphics and table formats are subject to change.

3. The temperature and wind speed are analyzed in this paper. What are the other influencing factors and how to control them?

4. The physical meaning of formulas and quantities can be written clearly.

5. The authors are suggested to discuss some recent works about lasers source, (i.e. Optics & Laser Technology 146, 107546, 2022; Ultrafast Science, 2022, 9870325, 6, 2022; Ultrafast Science, 2022, Article 9895418, 51, 2022; Ultrafast Science, 2022, 9893418, 9, 2022; Phys. Rev. Lett., 121, 023905 2018; Laser Photon. Rev. 13, 1800333, 2019)

Author Response

We appreciate valuable comments from reviewers. According to the comments, the manuscripts were refined. In addition, the manuscripts were checked and modified by the English-native-proofreading. The followings are responses to reviewers’ comments.

Reviewer 2 Report

In this study authors developed a new temperature and velocity measurement method that can measure three-dimensional temperature and flow velocity. I look forward to seeing it in print following some major revision.

1)      Introduction need to be improved. In introduction, authors discussed about two methods that other scholars developed to measure velocity and temperature simultaneously; however, it should be noted that such techniques are among the visualization techniques and your paper is just to measure the flow characteristics, not visualize them. In addition, there are other scholars that currently try to combine phosphor thermometry with PIV to obtain temperature and velocity simultaneously. Please also add such work in the introduction.

2)      On page 75 it is written that sensor part is 2 m long with a fiber diameter of approximately 0.15 mm. Authors should add a discussion on the mentioned range.

3)      in table 3, please modify R2 to R2

4)      What are the limitation of this technique, is this technique applicable on any flow velocities or temperature values? Please add more details about your proposed technique in abstract and conclusion.

5)      Please put all units of manuscript in Square brackets.

Author Response

(The authors gave the same response as above.)

Round 2

Reviewer 2 Report

Thank You all my concerns were answered.